# Resilience of S309 and AZD7442 monoclonal antibody treatments against infection by SARS-CoV-2 Omicron lineage strains

James Brett Case [1], Samantha Mackin[1,2], John M. Errico [2], Zhenlu Chong[1], Emily A. Madden [1], Bradley Whitener[1], Barbara Guarino[3], Michael A. Schmid [3], Kim Rosenthal[4], Kuishu Ren[4], Ha V. Dang [5], Gyorgy Snell[5], Ana Jung[2], Lindsay Droit[2], Scott A. Handley [2], Peter J. Halfmann [6], Yoshihiro Kawaoka [6,7,8], James E. Crowe Jr. [9,10,11], Daved H. Fremont [2,12,13], Herbert W. Virgin [2,5,14], Yueh-Ming Loo [4], Mark T. Esser [4], Lisa A. Purcell[5], Davide Corti [3] & Michael S. Diamond [1,2,12,15,16✉]

Omicron variant strains encode large numbers of changes in the spike protein compared to historical SARS-CoV-2 isolates. Although in vitro studies have suggested that several monoclonal antibody therapies lose neutralizing activity against Omicron variants, the effects in vivo remain largely unknown. Here, we report on the protective efficacy against three SARS-CoV-2 Omicron lineage strains (BA.1, BA.1.1, and BA.2) of two monoclonal antibody therapeutics (S309 [Vir Biotechnology] monotherapy and AZD7442 [AstraZeneca] combination), which correspond to ones used to treat or prevent SARS-CoV-2 infections in humans. Despite losses in neutralization potency in cell culture, S309 or AZD7442 treatments reduced BA.1, BA.1.1, and BA.2 lung infection in susceptible mice that express human ACE2 (K18-hACE2) in prophylactic and therapeutic settings. Correlation analyses between in vitro neutralizing activity and reductions in viral burden in K18-hACE2 or human FcγR transgenic mice suggest that S309 and AZD7442 have different mechanisms of protection against Omicron variants, with S309 utilizing Fc effector function interactions and AZD7442 acting principally by direct neutralization. Our data in mice demonstrate the resilience of S309 and AZD7442 mAbs against emerging SARS-CoV-2 variant strains and provide insight into the relationship between loss of antibody neutralization potency and retained protection in vivo.

[1] Department of Medicine, Washington University School of Medicine, St. Louis, MO, USA. [2] Department of Pathology & Immunology, Washington University School of Medicine, St. Louis, MO, USA. [3] Humabs BioMed SA, a subsidiary of Vir Biotechnology, Bellinzona, Switzerland. [4] Vaccines and Immune Therapies, BioPharmaceuticals R&D, AstraZeneca, Gaithersburg, MD, USA. [5] Vir Biotechnology, San Francisco, CA, USA. [6] Influenza Research Institute, Department of Pathobiological Sciences, School of Veterinary Medicine, University of Wisconsin-Madison, Madison, WI, USA. [7] Division of Virology, Institute of Medical Science, University of Tokyo, Tokyo, Japan. [8] The Research Center for Global Viral Diseases, National Center for Global Health and Medicine Research Institute, Tokyo, Japan. [9] Vanderbilt Vaccine Center, Vanderbilt University Medical Center, Nashville, TN, USA. [10] Department of Pediatrics, Vanderbilt University Medical Center, Nashville, TN, USA. [11] Department of Pathology, Microbiology, and Immunology, Vanderbilt University Medical Center, Nashville, TN, USA. [12] Department of Molecular Microbiology, Washington University School of Medicine, St. Louis, MO, USA. [13] Department of Biochemistry & Molecular Biophysics, Washington University School of Medicine, St. Louis, MO, USA. [14] University of Texas Southwestern Medical Center, Dallas, TX, USA. [15] Andrew M. and Jane M. Bursky Center for Human Immunology and Immunotherapy Programs, Washington University School of Medicine, Saint Louis, MO, USA. [16] Center for Vaccines and Immunity to Microbial Pathogens, Washington University School of Medicine, Saint Louis, MO, USA. ✉email: mdiamond@wustl.edu

Severe acute respiratory syndrome coronavirus 2 (SARS-CoV-2) variant strains continue to emerge and spread globally despite currently employed countermeasures and public health mandates. Since late 2020, variants of concern (VOC) and interest (VOI) have arisen due to continued SARS-CoV-2 evolution. Many variants contain substitutions in the N-terminal domain (NTD) and the receptor binding motif (RBM) of the receptor binding domain (RBD). Omicron lineage variants containing the largest numbers of spike protein changes described so far have emerged, spread globally, and become dominant. Moreover, cell-based studies suggest that the neutralizing activity of many monoclonal antibodies (mAbs) with Emergency Use Authorization (EUA) status or in advanced clinical development is diminished or abolished against Omicron lineage strains[1–4]. However, the effect of mutations that compromise antibody neutralization on their efficacy in vivo against SARS-CoV-2 remains less clear. Indeed, for some classes of broadly neutralizing mAbs against influenza[5,6] and Ebola[7,8] viruses, there is no strict correlation between neutralizing activity in vitro and protection in animal models.

Here, using mAbs that are currently in use to prevent or treat SARS-CoV-2 infection, we evaluate how the antigenic shift in Omicron viruses affects neutralization in cells and protection in mice. Despite some losses in neutralization potency against Omicron strains, we show that S309 and AZD7442 reduce viral burden and lung inflammation and thus retain appreciable in vivo activity against the Omicron variants tested. Correlation analyses between in vitro neutralization activity and in vivo reductions in lung infection suggest differing mechanisms of action for S309 and AZD7442, which we establish using genetic mAb variants and in vitro and in vivo assays.

## Results

**MAb neutralization against Omicron lineage viruses.** We analyzed the substitutions in the RBDs of BA.1 (B.1.1.529), BA.1.1 (B.1.1.529 R346K), and BA.2 strains (Fig. 1a, Supplementary Fig. 1) in the context of the structurally-defined binding epitopes of S309, a cross-reactive SARS-CoV mAb and the parent of sotrovimab [VIR-7831], and AZD8895 (tixagevimab) and AZD1061 (cilgavimab), two mAbs that together (AZD7442) form the clinically-used Evusheld combination treatment (Fig. 1b–e, Supplementary Fig. 1). S309 binds a conserved epitope on the RBD that is spatially distinct from the RBM[9] and the AZD8895 and AZD1061 antibodies bind non-overlapping RBM epitopes[10]. Across Omicron lineage strains, substitutions at several antibody contact residues have occurred (**S309**: G339D, R346K, N440K; **AZD8895**: K417N, S477N, T478K, E484A, Q493R; **AZD1061**: R346K, N440K, E484A, Q493R).

Because of these sequence changes, we assessed the neutralizing activity of S309, AZD8895, AZD1061, and AZD7442 against BA.1, BA.1.1, and BA.2 viruses in Vero-TMPRSS2 cells. For these studies, we used mAbs that correspond to the products in clinical use which have Fc modifications: S309-LS [M428L/N434S], AZD8895-YTE/TM [M252Y/S254T/T256E and L234F/L235E/P331S], AZD1061-YTE/TM, and AZD7442-YTE/TM. The LS and YTE Fc substitutions result in extended antibody half-life in humans, and the TM changes reduce Fc effector functions[11]. Compared to the historical WA1/2020 D614G strain (hereafter D614G), antibody incubation with BA.1 was associated with 2.5-fold (S309-LS), 25-fold (AZD7442-YTE/TM), 118-fold (AZD8895-YTE/TM), and 206-fold (AZD1061-YTE/TM) reductions in neutralization potency (Fig. 1f–o), which agree with experiments with authentic or pseudotyped SARS-CoV-2[1–4]. Some differences were observed with BA.1.1: whereas S309-LS and AZD8895-YTE/TM were only slightly less effective against

BA.1.1 compared to BA.1, the neutralizing activity of AZD1061-YTE/TM was reduced by almost 1,700-fold. Despite the decrease in activity of the AZD1061-YTE/TM component, the AZD7442-YTE/TM combination still showed inhibitory activity against BA.1.1 with a 176-fold reduction compared to D614G. Whereas small (no change to 5-fold) reductions in neutralization activity were observed with AZD1061-YTE/TM and AZD7442-YTE/TM against BA.2, larger reductions (32- and 68-fold) were observed for S309-LS and AZD8895-YTE/TM compared to D614G. We also observed lower binding affinity of S309, AZD8895, or AZD1061 Fab fragments to Omicron lineage RBDs, with the exception of AZD1061 and BA.2 (Supplementary Fig. 2, 3), which is consistent with neutralization trends for each mAb. Overall, these data demonstrate that S309 retains potency against BA.1 and BA.1.1 strains but has less in vitro neutralizing activity against BA.2, and the AZD7442 combination shows reduced yet residual activity against strains from all three Omicron lineages.

**MAb protection in vivo against Omicron viruses.** Because S309 and AZD7442 mAbs might act in vivo by a combination of mechanisms that are not fully reflected by in vitro neutralization potency assays, we evaluated the effects of the mutations observed in BA.1, BA.1.1 and BA.2 on efficacy in animals. For these studies, we used S309-LS and a different form of AZD7442, which contained only the TM substitutions and not the YTE modification but retains equivalent neutralizing activity[2]. Although the YTE modification promotes antibody recycling to confer extended antibody half-life in humans and non-human primates, it accelerates antibody elimination in rodents[12]. To assess the efficacy of S309-LS and AZD7442-TM in vivo, we administered a single 200 μg (~10 mg/kg total) mAb dose to K18-hACE2 transgenic mice by intraperitoneal injection one day prior to intranasal inoculation with BA.1, BA.1.1, or BA.2 strains. The clinical dosing in humans for sotrovimab is 500 mg and for AZD7442 is 300 mg of tixagevimab (AZD8895) + 300 mg of cilgavimab (AZD1061). Although Omicron lineage viruses are less pathogenic in mice, they still replicate to high levels in the lungs of K18-hACE2 mice[13]. Nonetheless, as preliminary studies suggested slightly different kinetics of replication and spread in mice, we harvested samples at 7 dpi for BA.1 and BA.1.1 and 6 dpi for BA.2[13]. In BA.1 and BA.1.1-infected mice, S309-LS mAb reduced viral burden in the lung, nasal turbinates, and nasal washes at 7 dpi compared to isotype mAb-control treated mice (Fig. 2a–d). However, control of infection by S309-LS, as judged by viral RNA levels, was less efficient against BA.1 (182-fold reduction) and BA.1.1 (39-fold reduction) viruses than against D614G (>500,000-fold reduction) (Supplementary Table 1). Despite the diminished neutralizing activity against BA.2 in vitro, S309-LS treatment reduced viral RNA levels in the lungs of BA.2-infected mice substantially (742-fold reduction) (Fig. 2a, b, Supplementary Table 1). Protection by S309-LS was not observed in the nasal turbinates or nasal washes of mice challenged with BA.2 (Fig. 2c, d), in part due to the low and variable levels of infection with this variant. AZD7442-TM treatment differentially reduced viral burden in the lungs of mice against D614G (>400,000-fold reduction in viral RNA), BA.1 (92-fold reduction in viral RNA), BA.1.1 (4-fold reduction in viral RNA), and BA.2 (>100,000-fold reduction in viral RNA) (Fig. 2e, f, Supplementary Table 1). Protection in the upper respiratory tract was less consistent, as AZD7442-TM treatment lowered viral RNA levels in the nasal washes of D614G and BA.1-infected mice but not in BA.1.1 or BA.2-infected mice and failed to reduce D614G, BA.1, BA.1.1, or BA.2 infection in the nasal turbinates (Fig. 2g, h).

As independent metrics of mAb protection, we measured cytokine and chemokine levels in lung homogenates and

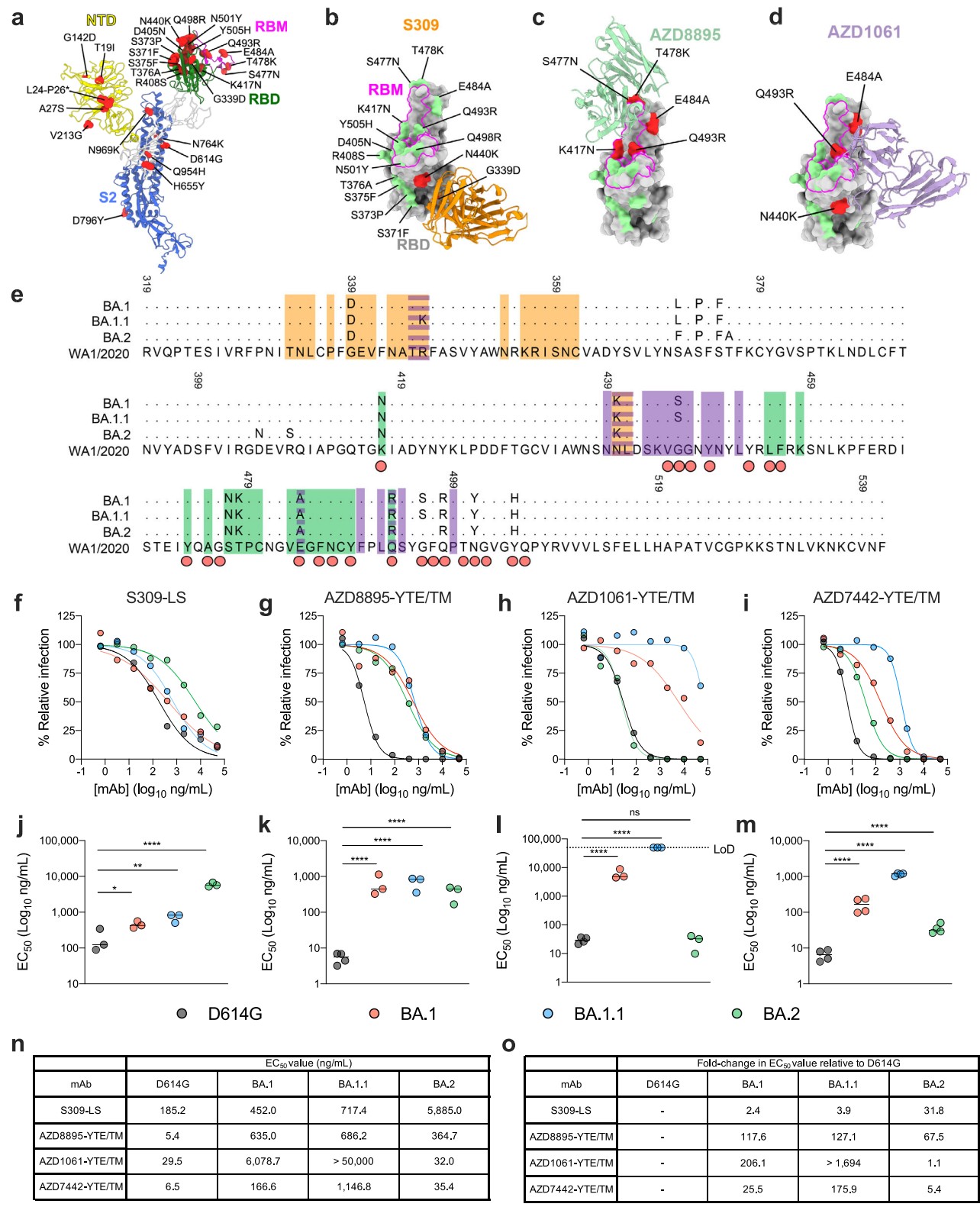

**Fig. 2** (partial, inferred from figure)

| n | EC50 value (ng/mL) | | | |
|---|---|---|---|---|
| mAb | D614G | BA.1 | BA.1.1 | BA.2 |
| S309-LS | 185.2 | 452.0 | 717.4 | 5,885.0 |
| AZD8895-YTE/TM | 5.4 | 635.0 | 686.2 | 364.7 |
| AZD1061-YTE/TM | 29.5 | 6,078.7 | > 50,000 | 32.0 |
| AZD7442-YTE/TM | 6.5 | 166.6 | 1,146.8 | 35.4 |

| o | Fold-change in EC50 value relative to D614G | | | |
|---|---|---|---|---|
| mAb | D614G | BA.1 | BA.1.1 | BA.2 |
| S309-LS | - | 2.4 | 3.9 | 31.8 |
| AZD8895-YTE/TM | - | 117.6 | 127.1 | 67.5 |
| AZD1061-YTE/TM | - | 206.1 | > 1,694 | 1.1 |
| AZD7442-YTE/TM | - | 25.5 | 175.9 | 5.4 |

analyzed lung sections for pathology from S309-LS and AZD7442-TM treated animals infected with Omicron variant strains (Fig. 2i, j, Supplementary Figs. 4–6). All infected K18-hACE2 mice receiving isotype control mAbs had increased expression levels of several pro-inflammatory cytokines and chemokines such as G-CSF, GM-CSF, IFN-γ, IL-1β, IL-6, CXCL-10, CCL-2, and CCL-4 when compared to uninfected mice. In contrast, mice treated with AZD7442-TM mAbs and infected with BA.1 or BA.2 but not BA.1.1. showed reduced levels of pro-inflammatory cytokines and chemokines, which is consistent with effects on viral burden (Fig. 2e, f). Compared to the isotype controls, mice treated with S309-LS had lower levels of cytokines and chemokines in lung homogenates after infection with all three Omicron variants, although the protection against BA.2-induced inflammation was less than against BA.1. or BA.1.1.

**Fig. 1 Neutralization of Omicron lineage strains by mAbs. a** One protomer of the SARS-CoV-2 spike trimer (PDB: 7C2L) is depicted with BA.2 variant amino acid substitutions labelled and shown as red spheres. The N-terminal domain (NTD), RBD, RBM, and S2 are colored in yellow, green, magenta, and blue, respectively. All mutated residues in the BA.2 RBD relative to WA1/2020 are indicated in **b**, and the BA.2 RBD bound by mAbs S309 (orange, PDB: 6WPS) (**b**), AZD8895 (green, PDB: 7L7D) (**c**), and AZD1061 (purple, PDB:7L7E) (**d**) are shown. BA.2 mutations in the respective epitopes of each mAb are shaded red, whereas those outside the epitope are shaded green. **e** Multiple sequence alignment showing the epitope footprints of each mAb on the SARS-CoV-2 RBD (orange, S309; green, AZD8895; purple, AZD1061). The WA1/2020 RBD is shown in the last row with relative variant sequence changes indicated. Red circles below the sequence alignment indicate hACE2 contact residues on the SARS-CoV-2 RBD[43]. Structural analysis and depictions were generated using UCSF ChimeraX[44]. **f–i** Neutralization curves in Vero-TMPRSS2 cells with the indicated SARS-CoV-2 strain and mAb. The average of three to four experiments performed in technical duplicate are shown. **j–m** Comparison of $EC_{50}$ values for the indicated mAb against D614G, BA.1, BA.1.1, and BA.2 viruses. Data are the average of three experiments, error bars indicate standard error of the mean (SEM), and the dashed line indicates the upper limit of detection (one-way ANOVA with Dunnett's test; ns, not significant, *$P < 0.05$, **$P < 0.01$, ***$P < 0.001$, ****$P < 0.0001$). **n** Summary of the $EC_{50}$ values for each mAb against the indicated SARS-CoV-2 strain. **o** Summary of the fold-change in $EC_{50}$ values for each mAb against the indicated Omicron strain relative to SARS-CoV-2 D614G. Source data are provided as a Source Data file.

Histopathological analysis of lungs from isotype-treated, but not S309-LS- or AZD7442-TM-treated, D614G-infected K18-hACE2 mice at 7 dpi showed evidence of pneumonia with immune cell infiltration, alveolar space consolidation, and edema (Supplementary Fig. 6). Although infection of rodents with BA.1, BA.1.1, or BA.2 strains results in less pathogenesis than D614G strains[13–16], focal pneumonia still was observed in isotype control mAb-treated, Omicron strain-infected K18-hACE2 mice. In comparison, S309-LS or AZD7442-TM treatment prevented immune cell infiltration and airspace consolidation. Overall, these experiments suggest that despite losses in neutralizing potency in cell culture, S309-LS or AZD7442-TM treatment can limit inflammation and pathogenesis in the lung caused by Omicron variants.

We next evaluated whether the differences in neutralizing activity of S309-LS and AZD7442-YTE/TM correlated with changes in lung viral RNA levels after infection with the three Omicron strains. The change in AZD7442-YTE/TM neutralizing activity associated directly with the differences in lung viral burden of each Omicron variant (Fig. 2k). This relationship is consistent with its likely mechanism of action, virus neutralization and inhibition of entry[17,18]. The AZD7442-TM version we used, like the clinical drug tixagevimab + cilgavimab, encodes for modifications in the constant region of the mAb heavy chains that profoundly decrease binding to Fc-gamma receptors (FcγRs) and complement components (ref. [19] and Fig. 3a). In comparison, for S309-LS, a similar direct correlation between changes in neutralization potency in vitro and reductions in viral burden in vivo was not observed (Fig. 2l), indicating a possible additional protective mechanism beyond virus neutralization.

**S309-LS employs Fc effector functions to protect against SARS-CoV-2 variants.** To evaluate a potential role for Fc effector functions in S309 mAb-mediated protection against Omicron strains, we engineered loss-of-function GRLR mutations (G236R, L328R) into the Fc domain of the human IgG1 heavy chain of S309; these substitutions eliminate antibody binding to FcγRs[11]. Introduction of the GRLR mutations abrogated binding to hFcγRI, hFcγRIIIa, and mFcγRIV, as expected (Fig. 3a) but did not affect neutralization of the SARS-CoV-2 strains (Supplementary Fig. 7). Next, we compared VIR-7831 (the clinical form of S309-LS) and S309-GRLR in an in vitro antibody-dependent cell cytotoxicity (ADCC) assay. When target cells expressing similar levels of Wuhan-D614, BA.1, or BA.2 spike proteins on the cell surface were incubated with VIR-7831 mAb, we observed some reductions in binding to Omicron spike proteins compared to mAb S2X324 (Fig. 3b, c), an antibody that retains neutralizing activity against BA.1, BA.1.1, and BA.2 and engages a distinct epitope in the RBM[20]. Despite the differences in binding, target cells expressing Wuhan-D614, BA.1, or BA.2 spike proteins were lysed efficiently by primary natural killer (NK) cells

(antibody-dependent cellular cytotoxicity, ADCC) isolated from four donors by VIR-7831 but not by S309-GRLR (Fig. 3d, e, Supplementary Fig. 8a). Similarly, primary CD14[+] monocytes isolated from five donors mediated comparable antibody-dependent cellular phagocytosis (ADCP) of target cells expressing Wuhan-D614, BA.1, or BA.2 spike proteins by VIR-7831 but not by S309-GRLR (Fig. 3f, g, Supplementary Fig. 8b, c).

To evaluate the role of effector functions in vivo in S309-LS mAb-mediated protection against Omicron variant strains, we treated K18-hACE2 mice with a single 200 μg (~10 mg/kg total) dose of S309-GRLR mAb by intraperitoneal injection one day prior to intranasal inoculation with D614G, BA.1, or BA.2 strains. At 6 (BA.2) or 7 (D614G and BA.1) dpi, viral RNA levels in the lungs, nasal turbinates, and nasal washes were measured (Fig. 4a–c, Supplementary Table 1). Although S309-GRLR treatment reduced viral burden in the lung (6-fold) and nasal turbinates (10-fold) of D614G-infected mice, reductions were modest compared to mice receiving S309-LS (>500,000-fold and 230-fold respectively; see Fig. 2a, c, Supplementary Table 1); moreover, S309-GRLR did not limit infection by BA.1 and BA.2 strains in the tissues tested. Indeed, similarly high levels of pro-inflammatory cytokine and chemokine were measured in the lung homogenates of S309-GRLR and isotype control mAb-treated mice infected with D614G and BA.1 strains (Fig. 4d, Supplementary Fig. 9). To corroborate these findings, we treated hFcγR transgenic C57BL/6 mice[21] (hFcγR Tg) with a single 3 mg/kg dose of S309-LS or S309-GRLR mAbs one day prior to inoculation with a SARS-CoV-2 Beta (B.1.351) isolate; we used the Beta isolate for these studies because Omicron strains replicate poorly in conventional C57BL/6 mice lacking expression of hACE2[13]. At 2 or 4 dpi, S309-LS mAb-treated hFcγR Tg mice showed markedly reduced levels of viral RNA (15 to 47 fold or infectious virus 81- to 292-fold) in the lung compared to the isotype control mAb-treated mice, whereas animals administered S309-GRLR showed smaller (3-fold) differences, most of which did not attain statistical significance (Fig. 4e, f, Supplementary Table 1). Collectively, these data suggest that the protection mediated by S309-LS mAb in vivo relies to a large extent on Fc effector functions and engagement of FcγRs.

We next evaluated the therapeutic potential of S309-LS and AZD7442-TM against D614G and each of the Omicron variants by administering mAbs at one day post-virus inoculation. Both S309-LS and AZD7442-TM reduced viral RNA titers in the lungs and nasal turbinates (Fig. 4g, h and Supplementary Table 1) against all variants tested, with the one exception (AZD7442-TM versus BA.1.1 in the nasal turbinates). Moreover, therapeutic administration of S309-LS and AZD7442-TM reduced viral RNA levels in the nasal washes of D614G, BA.1, and BA.1.1-infected, but not BA.2-infected, mice (Fig. 4i). Overall, these data establish that S309-LS and AZD7442-TM have post-exposure therapeutic activity against multiple Omicron variant strains.

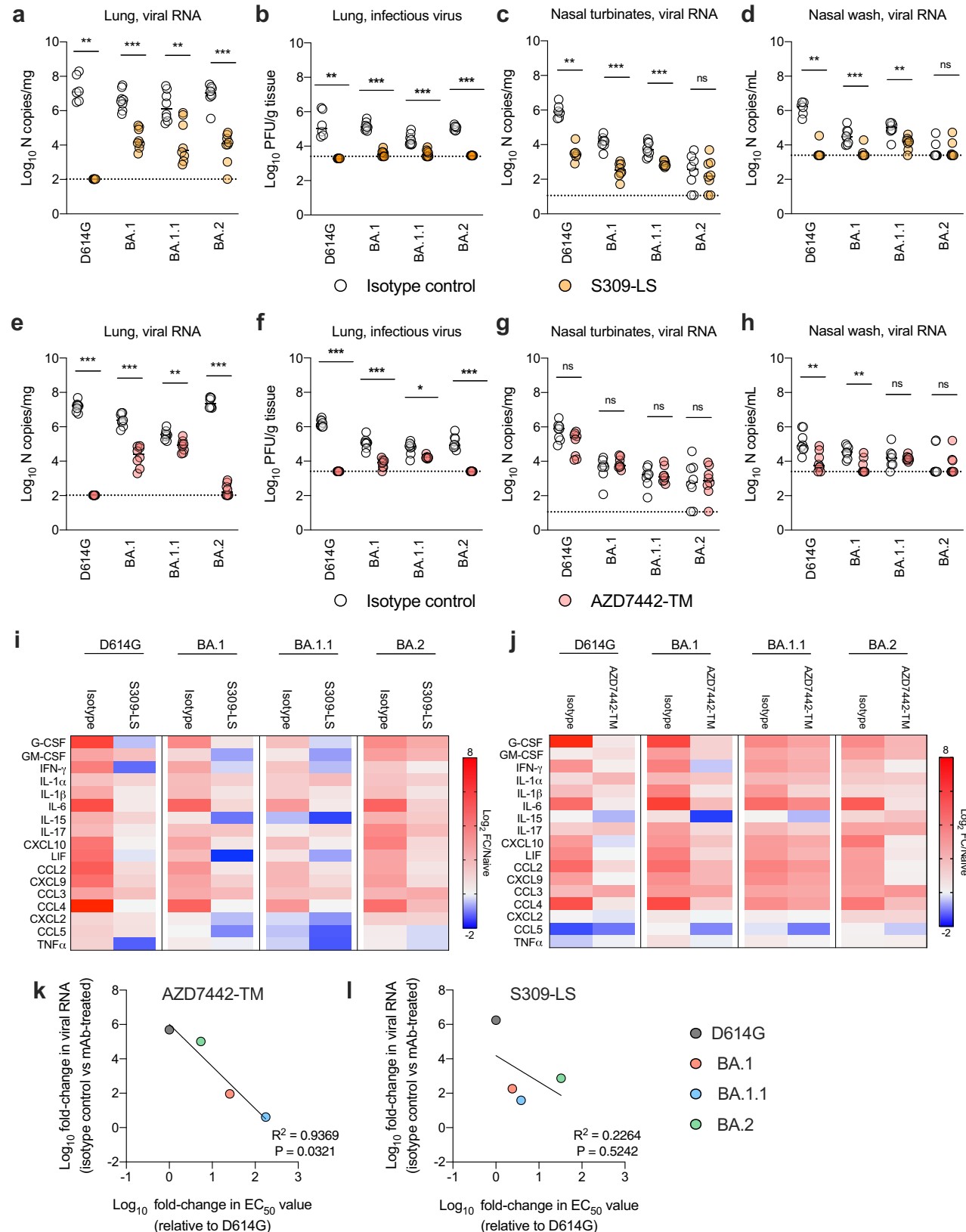

## Discussion

Due to the continued emergence of SARS-CoV-2 variants encoding an increasing number of amino acid changes in the spike protein, antibody countermeasure efficacy requires continued monitoring. When the BA.1 Omicron virus emerged in late 2021, five mAb therapies were in late-stage clinical development or had acquired EUA status. In vitro assays with pseudoviruses[1] and authentic viruses[2] established that mAb therapies from Regeneron (REGN10933 and REGN10987), Lilly (LY-CoV555 and LY-CoV016), and Celltrion (CT-P59) showed a complete loss in neutralizing activity against BA.1. Subsequent experiments in K18-hACE2 mice confirmed that the REGN-

**Fig. 2 Antibody protection against Omicron variants in K18-hACE2 mice. a–j** Eight-week-old female K18-hACE2 mice received 200 µg (about 10 mg/kg) of the indicated mAb treatment by intraperitoneal injection one day before intranasal inoculation with $10^3$ FFU of the indicated SARS-CoV-2 strain. Tissues were collected at six (BA.2) or seven days (all other strains) after inoculation. Viral RNA levels in the lungs (**a**, **e**), nasal turbinates (**c**, **g**), and nasal washes (**d**, **h**) were determined by RT-qPCR, and infectious virus in the lungs (**b**, **f**) was assayed by plaque assay (lines indicate median ± SEM.; $n = 6$–8 mice per group, two experiments; Two-tailed Mann-Whitney test between isotype and mAb treatment; ns, not significant; $^*P < 0.05$, $^{**}P < 0.01$, $^{***}P < 0.001$). **i**, **j** Heat map of cytokine and chemokine protein expression levels in lung homogenates from the indicated groups. Data are presented as $\log_2$-transformed fold-change over naive mice. Blue, reduction; red, increase. **k**, **l**, Correlation analysis. The fold-change in $EC_{50}$ value of AZD7442-YTE/TM (**k**) and S309-LS (**l**) for D614G and each Omicron variant strain are plotted on the x-axis. The fold-change in lung viral RNA titer between the respective isotype or mAb-treated groups against each Omicron variant strain are plotted on the y-axis. Best-fit lines were calculated using a simple linear regression. Two-tailed Pearson correlation was used to calculate the $R^2$ and P values indicated within each panel. Source data are provided as a Source Data file.

COV2 mAb cocktail completely lost its efficacy against the BA.1 variant[22]. More recently, an additional antibody (LY-CoV1404, bebtelovimab), which shows considerable neutralization activity against a range of SARS-CoV-2 strains, received EUA status[23], although protection data in vivo against VOC, including Omicron, has not yet been published. Our data suggest that both neutralizing and Fc effector activities of mAb therapies should be considered when making recommendations about dosing and usage against emerging SARS-CoV-2 variants.

We compared the in vitro neutralizing activity and in vivo efficacy of S309 (parent mAb of sotrovimab) and AZD7442 (tixagevimab + cilgavimab) that correspond to the clinically-used products. Our study establishes the utility of S309 and AZD7442 mAbs against highly divergent SARS-CoV-2 variants. Despite losses in neutralization potency against BA.1, BA.1.1, and BA.2 strains, S309-LS and AZD7442-TM reduced viral burden and pro-inflammatory cytokine levels in the lungs of K18-hACE2 mice, albeit with some differences in activity and mechanisms of action. Although AZD7442-TM had a limited protective effect on viral burden in the nasal washes and nasal turbinates of infected mice, this was not entirely unexpected, as studies with the parental mAbs COV2-2196 and COV2-2130 showed less protection in nasal washes than lungs against multiple SARS-CoV-2 VOC[24]. Moreover, studies in non-human primates with anti-SARS-CoV-2 human mAbs showed the concentrations in nasopharyngeal washes are approximately 0.1% of those found in the serum[25], which likely explains their diminished benefit in this tissue compartment.

We also assessed whether the reductions in mAb neutralization potency against Omicron variant strains correlated with the observed changes in viral burden. For AZD7442-TM, which contains L234F/L235E/P331S modifications that abolish Fc receptor engagement[11] and were introduced to decrease the potential risk of antibody-dependent enhancement of disease[18], antibody-mediated reductions in viral titer correlated directly with neutralization activity against Omicron variant strains; thus, neutralization is likely the key protective mechanism for these RBM-specific mAbs. For S309-LS, which only contains half-life extending M428L/N434S modifications in the human IgG1 Fc domain, and exhibits Fc effector functions including ADCC and ADCP[9], changes in neutralization potency did not linearly relate to changes in lung viral titer. S309-LS mAb treatment still conferred significant protection in the lungs of mice infected with BA.2 despite a substantial loss in neutralizing activity. Because of these results, we evaluated the contributions of Fc effector functions in protection in mice using S309-GRLR, which has G236R/L328R mutations in the Fc domain that abrogate binding to FcγRs[11]. We observed that intact S309-LS but not S309-GRLR mAb protected K18-hACE2 and hFcγR Tg mice against SARS-CoV-2 variant strains. These results are consistent with prior studies showing a beneficial role of Fc-effector functions in antibody mediated protection in mice and hamsters[26–30], and may explain why mAbs with markedly different in vitro neutralization potencies against SARS-CoV-2 strains show similar

protective activity in animals (https://opendata.ncats.nih.gov/covid19/animal). Furthermore, they also demonstrate that for some mAbs, Fc effector functions can compensate for losses in neutralization potency against SARS-CoV-2 variants and act as a protective mechanism in vivo. Thus, effector functions can contribute to resilience of some mAbs against Omicron and other VOC[31,32]. We speculate that the stoichiometric threshold and antibody occupancy requirements for Fc effector function activity may be lower than for virion neutralization[33]; if so, this property might clarify how antibodies with reduced neutralizing potency against VOC that still bind spike protein on the virion or surface of infected cells retain protective activity in vivo. Alternatively, potential differences in spike expression on the surface of Omicron (BA.1, BA.1.1, and BA.2)-infected cells could further modulate S309-Fc-mediated effector function activity.

We note several limitations of our study: (a) Female K18-hACE2 mice were used to allow for group caging. Follow-up experiments in male mice to confirm and extend these results are needed. (b) The BA.1, BA.1.1., and BA.2 viruses are less pathogenic in mice than the D614G virus[13–16]. This could lead to an overestimation of protection compared to other more virulent strains in mice. (c) The relationship between initial viral dosing and antibody protection against Omicron variants was not explored. (d) Several experiments were performed in transgenic mice that over-express human ACE2 receptors. High levels of cellular hACE2 can diminish the neutralizing activity of mAbs that bind non-RBM sites of the SARS-CoV-2 spike[34,35]. Thus, studies in hACE2-transgenic mice could underestimate the efficacy of mAbs binding outside of the RBM. Challenge studies in other animal models and ultimately humans will be required for corroboration. (e) We did not assess mAb efficacy against the newest emerging Omicron variants including BA.2.11, BA.2.12.1, BA.4, or BA.5, which recently were linked to antibody escape in pseudovirus-based neutralization assays[36,37]. In vitro and in vivo evaluation of S309 (sotrovimab) and AZD7442 (tixagevimab + cilgavimab) against these strains using authentic SARS-CoV-2 viruses are needed. (f) We did not identify the specific immune effector cells that mediate the protective Fc-mediated effector responses in vivo. Future studies are needed to determine the cell types and inflammatory mediators responsible for this mechanism of antibody protection.

Collectively, our data expand on recent in vitro findings with BA.1 strains by evaluating the level of protection conferred by treatment with two EUA mAbs against the three currently dominant Omicron variants. While S309-LS (and by extension sotrovimab) and AZD7442-TM (tixagevimab + cilgavimab) retained inhibitory activity against several Omicron lineage strains, the impact of shifts in neutralization potency in vitro may not directly predict dosing in the clinical setting. Finally, our studies highlight the potential of both mAb neutralization and Fc effector function mechanisms in protecting against SARS-CoV-2-mediated disease and suggest mechanisms of action for withstanding mutations in variant strains that reduce but do not abrogate mAb binding and neutralization.

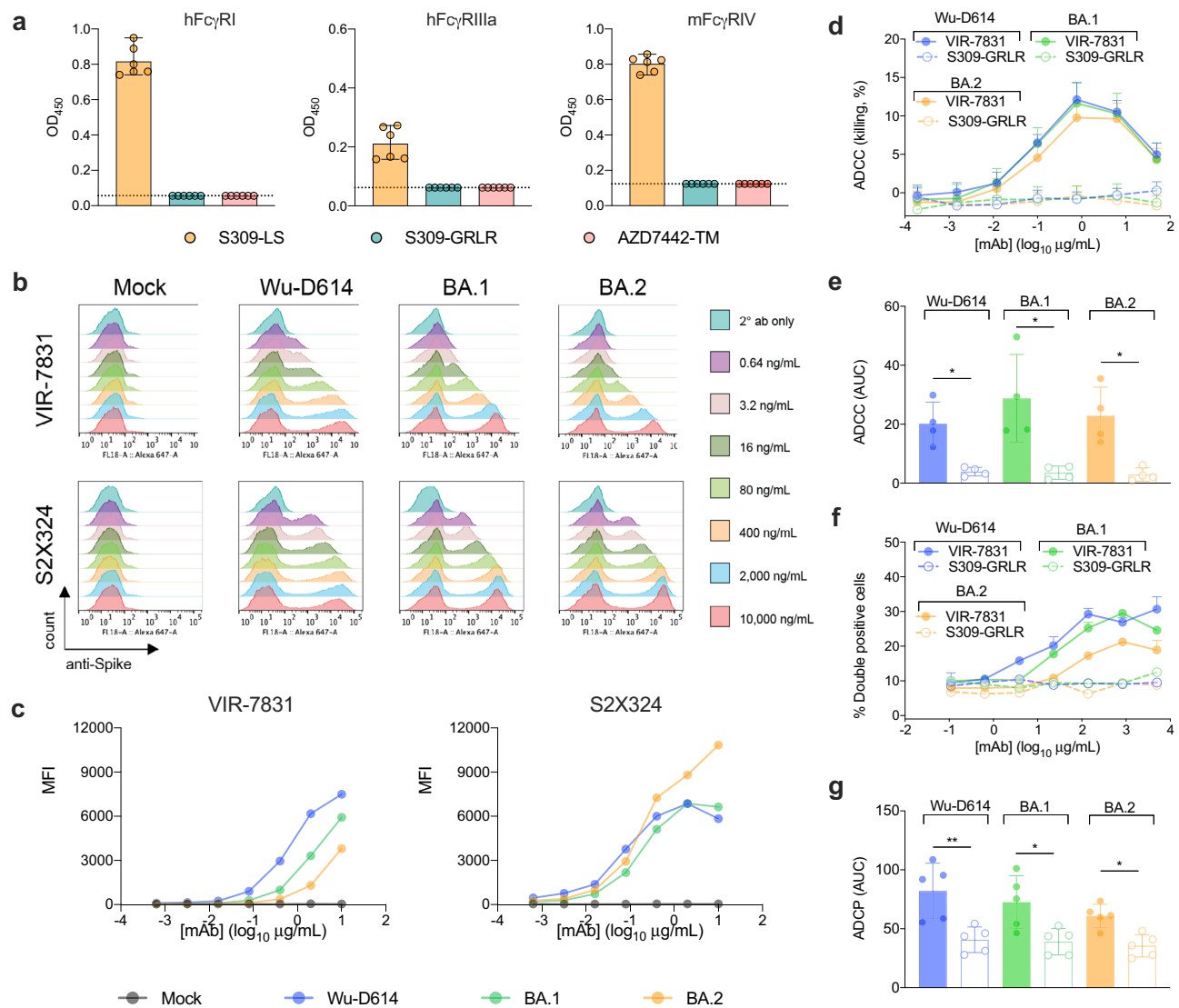

**Fig. 3 VIR-7831 binds and instructs effector cells for ADCC and ADCP. a** Binding of S309-LS, S309-GRLR, or AZD7442-TM mAbs to hFcγRI, hFcγRIIIa, or mFcγRIV (two experiments; dotted lines indicate the limit of detection; data are presented as meanvalues ± range). **b** ExpiCHO-S cells were transiently transfected with plasmids encoding the indicated SARS-CoV-2 spike protein. 24 to 48 h later, cells were harvested, washed, and stained with the indicated concentrations of VIR-7831 or S2X324 mAbs to assess binding to the cell surface. Representative histograms from two or three experiments are shown. **c** Antibody binding curves for VIR-7831 and S2X324 using the data in **b** and presented as mean fluorescence intensity (MFI) versus antibody concentration. **d**, **e** ExpiCHO-S cells transiently transfected with Wuhan-1 D614, BA.1, or BA.2 spike proteins were incubated with the indicated concentrations of VIR-7831 or S309-GRLR mAb and mixed with purified NK cells isolated from healthy donors at a ratio of 1:9 (target:effector). Cell lysis was determined by a lactate dehydrogenase release assay. Data are presented as mean values ± standard deviations (SD) from one representative of four donors (**d**). Area under the curve (AUC) analyses from four NK donors (**e**) (see Supplementary Fig. 8). **f**, **g** ExpiCHO-S cells transiently transfected with Wuhan-1 D614, BA.1, or BA.2 spike proteins and fluorescently labelled with PKH67 were incubated with the indicated concentrations of VIR-7831 or S309-GRLR mAb and mixed with PBMCs labelled with CellTrace Violet from healthy donors at a ratio of 1:20 (target:PBMCs). Association of CD14+ monocytes with spike-expressing target cells (ADCP) was determined by flow cytometry. Data are presented as mean values ± SD from one representative of five donors (**f**). AUC analyses of VIR-7831 and S309-GRLR for each Omicron variant for five donors (**g**). **e**, **g** Two-tailed Mann-Whitney test between VIR-7831 and S309-GRLR for the indicated variant; *$P < 0.05$, **$P < 0.01$. Source data are provided as a Source Data file. Gating strategies are shown in Supplementary Fig. 8c.

## Methods

**Cells**. Vero-TMPRSS2[38] and Vero-hACE2-TMPRRS2[39] cells were cultured at 37 °C in Dulbecco's Modified Eagle medium (DMEM) supplemented with 10% fetal bovine serum (FBS), 10 mM HEPES pH 7.3, 1 mM sodium pyruvate, 1× non-essential amino acids, and 100 U/ml of penicillin–streptomycin. Vero-TMPRSS2 cells were supplemented with 5 µg/mL of blasticidin. Vero-hACE2-TMPRSS2 cells were supplemented with 10 µg/mL of puromycin. ExpiCHO-S cells were obtained from Thermo Fisher Scientific. All cells routinely tested negative for mycoplasma using a PCR-based assay.

**Viruses**. The Beta (B.1.351) and Omicron (BA.1 (R346), BA.1.1 (R346K), and BA.2) strains were obtained from nasopharyngeal isolates. All virus stocks were

generated in Vero-TMPRSS2 cells and subjected to next-generation sequencing as described previously[39] to confirm the presence and stability of expected substitutions (see Supplementary Table 2). All virus experiments were performed in an approved biosafety level 3 (BSL-3) facility.

**Monoclonal antibody purification**. The mAbs studied in this paper, S309, AZD8895, AZD1061, and the AZD7442 cocktail have been described previously[9,18,20].

S309-LS and S309-GRLR were produced in ExpiCHO-S cells and affinity-purified using HiTrap Protein A columns (GE Healthcare, HiTrap mAb select Xtra #28-4082-61) followed by buffer exchange to histidine buffer (20 mM histidine, 8% sucrose, pH 6.0) using HiPrep 26/10 desalting columns. The final products were

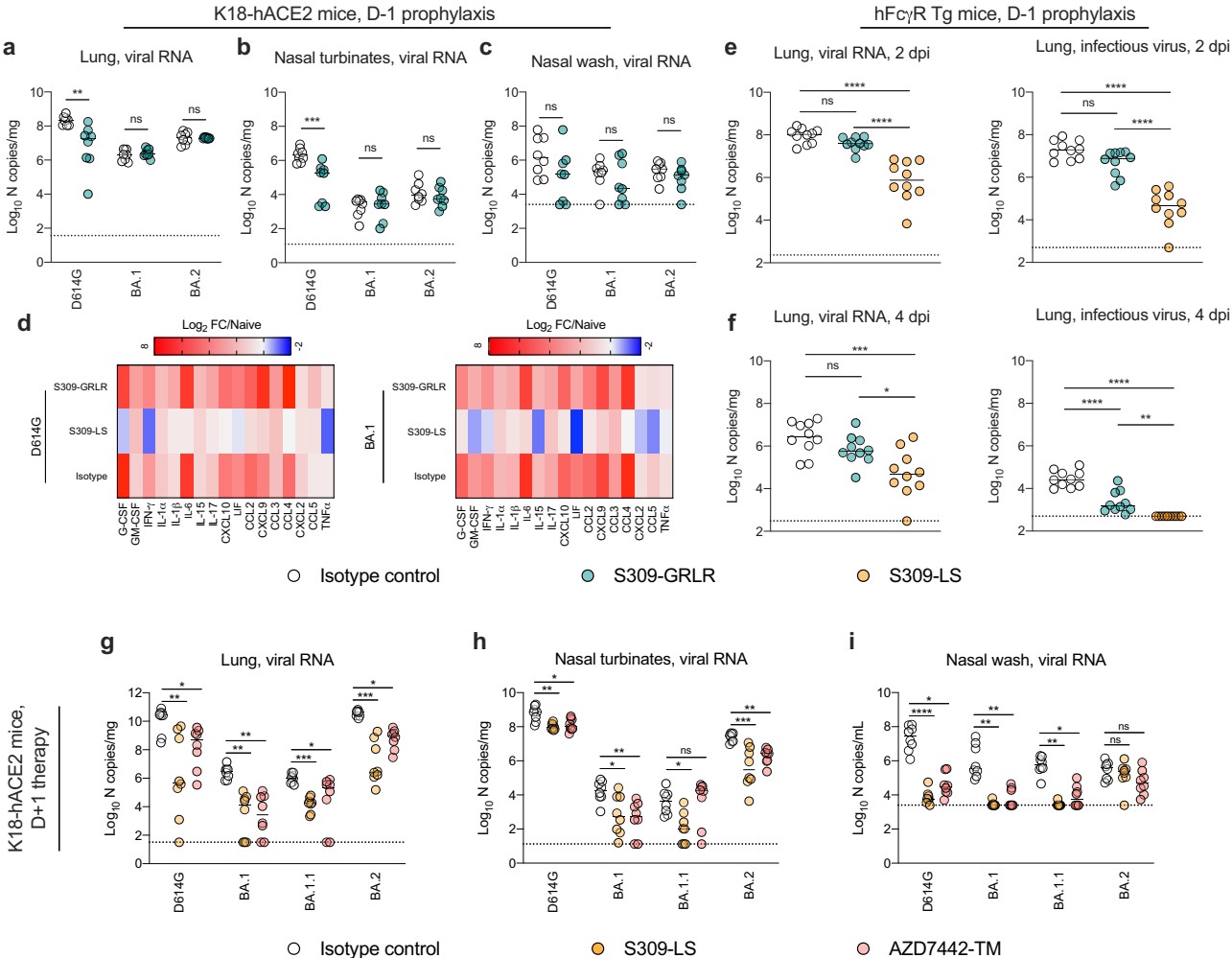

**Fig. 4 Fc-effector functions and mAb-mediated protection. a–d** Eight-week-old female K18-hACE2 mice or (**e**, **f**) 12-week-old male hFcγR Tg mice received a single 10 mg/kg or 3 mg/kg dose respectively, of isotype control, S309-LS or S309-GRLR mAb by intraperitoneal injection one day before intranasal inoculation with $10^3$ FFU of D614G, BA.1, or BA.2 (**a–c**) or $10^5$ FFU of Beta (B.1.351) (**e**, **f**). Tissues were collected at 2 (B.1.351), 4 (B.1.351), 6 (BA.2), or 7 (D614G and BA.1) dpi. Viral RNA levels in the lungs (**a**, **e**, **f**), nasal turbinates (**b**), and nasal washes (**c**) were determined by RT-qPCR, and infectious virus in the lungs (**e**, **f**) was measured by plaque assay. **a–c**, **e**, **f** lines indicate median ± SEM.; **a–c** and **e**, **f** $n = 8$ and 10 mice per group, respectively; two experiments; **a–c** Two-tailed Mann-Whitney test between isotype and mAb treatment; ns, not significant; \*\*$P < 0.01$, \*\*\*$P < 0.001$ **e**, **f** one-way ANOVA with Tukey's multiple comparisons test; ns, not significant; \*$P < 0.05$, \*\*$P < 0.01$, \*\*\*$P < 0.001$, \*\*\*\*$P < 0.0001$. **d** Heat map of cytokine and chemokine protein expression levels in lung homogenates from the indicated groups. Data are presented as $\log_2$-transformed fold-change over naive mice. Blue, reduction; red, increase. S309-LS data in Fig. 2i is included for comparison. **g–i** Eight-week-old female K18-hACE2 mice were inoculated with $10^3$ FFU of the indicated SARS-CoV-2 strain by intranasal administration one day before receiving a single 30 mg/kg dose of S309-LS or AZD7442-TM mAb by intraperitoneal injection. Tissues were collected at 6 (BA.2) or 7 (all other strains) dpi. Viral RNA levels in the lungs (**g**), nasal turbinates (**h**), and nasal washes (**i**) were determined by RT-qPCR. **g–i** lines indicate median ± SEM; $n = 8$ mice per group; two experiments; Kruskal-Wallis test between isotype control and each mAb treatment with Dunn's multiple comparisons test; ns, not significant; \*$P < 0.05$, \*\*$P < 0.01$, \*\*\*$P < 0.001$, \*\*\*\*$P < 0.0001$. Source data are provided as a Source Data file.

sterilized after passage through 0.22 μm filters and stored at 4 °C. VIR-7831 (clinical lead variant of S309-LS) was produced at WuXi Biologics.

AZD8895 and AZD1061 mAbs were cloned into mammalian expression vectors and expressed as IgG1 constructs with the TM (L234F/L235E/P331S) Fc modification with or without a second YTE (M252Y/S254T/T256E) modification to extend half-life in humans. MAbs were expressed in 293 F cells after transfection with 293fectin (Thermo Fisher Scientific) and isolated from supernatants by affinity chromatography using Protein A or Protein G columns (GE Healthcare). MAbs were eluted with 0.1 M glycine at low pH and dialyzed into PBS.

**Mouse experiments.** Animal studies were carried out in accordance with the recommendations in the Guide for the Care and Use of Laboratory Animals of the National Institutes of Health. The protocols were approved by the Institutional Animal Care and Use Committee at the Washington University School of Medicine (assurance number A3381–01). Virus inoculations were performed under

anesthesia that was induced and maintained with ketamine hydrochloride and xylazine, and all efforts were made to minimize animal suffering.

Heterozygous K18-hACE2 C57BL/6 J mice (strain: 2B6.Cg-Tg(K18-ACE2) 2Prlmn/J) were obtained from The Jackson Laboratory. Human FcγR Tg mice[21] (FcγRα$^{−/−}$/hFcγRI$^+$/hFcγRIIA$^{R131+}$/hFcγRIIB$^+$/hFcγRIIIA$^{F158+}$/hFcγRIIIB$^+$) were a generous gift (J. Ravetch, Rockefeller University) and bred at Washington University. All animals were housed in groups of 3 to 5 and fed standard chow diets. The photoperiod was 12 h on:12 h off dark/light cycle. The ambient animal room temperature was 70° F, controlled within ±2° and the room humidity was 50%, controlled within ±5%.

For experiments with K18-hACE2 mice, eight- to ten-week-old female mice were administered the indicated doses of the respective SARS-CoV-2 strains (see Figure legends) by intranasal administration. For hFcγR Tg mouse experiments, 12-week-old male mice were administered $10^5$ FFU of a Beta (B.1.351) isolate by intranasal administration. In vivo studies were not blinded, and mice were randomly assigned to treatment groups. No sample-size calculations were performed to power each study. Instead, sample sizes were determined based on prior in vivo virus challenge

experiments. Mice were administered the indicated mAb dose by intraperitoneal injection one day before or after intranasal inoculation with the indicated SARS-CoV-2 strain. AZD7442-TM (lacking the YTE modification that accelerates antibody elimination in rodents) was used in mouse studies.

**Focus reduction neutralization test**. Serial dilutions of mAbs were incubated with $10^2$ focus-forming units (FFU) of different strains or variants of SARS-CoV-2 for 1 h at 37 °C. Antibody-virus complexes were added to Vero-TMPRSS2 cell monolayers in 96-well plates and incubated at 37 °C for 1 h. Subsequently, cells were overlaid with 1% (w/v) methylcellulose in MEM. Plates were harvested 48–72 h later by removing overlays and fixing with 4% PFA in PBS for 20 min at room temperature. Plates were washed and incubated with an oligoclonal pool of SARS2-2, SARS2–11, SARS2–16, SARS2–31, SARS2–38, SARS2–57, and SARS2–71[40]. Plates with Omicron variant strains were additionally incubated with CR3022 and a pool of anti-SARS-CoV-2 mAbs that cross-react with SARS-CoV[41]. Subsequently, samples were incubated with HRP-conjugated goat anti-mouse IgG (Sigma, 12–349) and HRP-conjugated goat anti-human IgG (Sigma, A6029) in PBS supplemented with 0.1% saponin and 0.1% bovine serum albumin. SARS-CoV-2-infected cell foci were visualized using TrueBlue peroxidase substrate (KPL) and quantitated on an ImmunoSpot microanalyzer (Cellular Technologies).

**Measurement of viral RNA levels**. Tissues were weighed and homogenized with zirconia beads in a MagNA Lyser instrument (Roche Life Science) in 1 mL of DMEM medium supplemented with 2% heat-inactivated FBS. Tissue homogenates were clarified by centrifugation at approximately $10,000 \times g$ for 5 min and stored at −80 °C. RNA was extracted using the MagMax mirVana Total RNA isolation kit (Thermo Fisher Scientific) on the Kingfisher Flex extraction robot (Thermo Fisher Scientific). RNA was reverse transcribed and amplified using the TaqMan RNA-to-CT 1-Step Kit (Thermo Fisher Scientific). Reverse transcription was carried out at 48 °C for 15 min followed by 2 min at 95 °C. Amplification was accomplished over 50 cycles as follows: 95 °C for 15 s and 60 °C for 1 min. Copies of SARS-CoV-2 $N$ gene RNA in samples were determined using a previously published assay[42]. Briefly, a TaqMan assay was designed to target a highly conserved region of the $N$ gene (Forward primer: ATGCTGCAATCGTGCTACAA; Reverse primer: GACTGCCGCCTCTGCTC; Probe: /56-FAM/TCAAGGAAC/ZEN/AACATTGC-CAA/3IABkFQ/). This region was included in an RNA standard to allow for copy number determination down to 10 copies per reaction. The reaction mixture contained final concentrations of primers and probe of 500 and 100 nM, respectively.

**Viral plaque assay**. Vero-TMPRSS2-hACE2 cells were seeded at a density of $1 \times 10^5$ cells per well in 24-well tissue culture plates. The following day, medium was removed and replaced with 200 µL of material to be titrated diluted serially in DMEM supplemented with 2% FBS. One hour later, 1 mL of methylcellulose overlay was added. Plates were incubated for 72 h, then fixed with 4% paraformaldehyde (final concentration) in PBS for 20 min. Plates were stained with 0.05% (w/v) crystal violet in 20% methanol and washed twice with distilled, deionized water.

**Transient expression of recombinant SARS-CoV-2 protein and flow cytometry**. ExpiCHO-S cells were seeded at $6 \times 10^6$ cells/mL in a volume of 5 mL in a 50 mL bioreactor. The following day, cells were transfected with SARS-CoV-2 spike glycoprotein-encoding pcDNA3.1(+) plasmids (BetaCoV/Wuhan-Hu-1/2019, accession number MN908947, Wuhan D614; Omicron BA.1 and BA.2 generated by overlap PCR mutagenesis of the Wuhan D614 plasmid) harboring the Δ19 C-terminal truncation[27]. Spike encoding plasmids were diluted in cold OptiPRO SFM (Life Technologies, 12309-050), mixed with ExpiFectamine CHO Reagent (Life Technologies, A29130) and added to cells. Transfected cells were then incubated at 37˚C with 8% $CO_2$ with an orbital shaking speed of 250 RPM (orbital diameter of 25 mm) for 24 to 48 h. Transiently transfected ExpiCHO-S cells were harvested and washed twice in wash buffer (PBS 2% FBS, 2 mM EDTA). Cells were counted and distributed into round bottom 96-well plates (Corning, 3799) and incubated with serial dilutions of mAb starting at 10 µg/mL. Alexa Fluor647-labelled Goat Anti-human IgG secondary Ab (Jackson ImmunoResearch, 109–606–098) was prepared at 2 µg/mL and added onto cells after two washing steps. Cells were then washed twice and resuspended in wash buffer for data acquisition at ZE5 cytometer (BioRad).

**Fc-mediated effector functions**. Primary cells were collected from healthy human donors with informed consent and authorization via the *Comitato Etico Canton Ticino* (Switzerland). ADCC assays were performed using ExpiCHO-S cells transiently transfected with SARS-CoV-2 spike glycoproteins (Wuhan D614, BA.1 or BA.2) as targets. NK cells were isolated from fresh blood of healthy donors using the MACSxpress NK Isolation Kit (Miltenyi Biotec, cat. no. 130-098-185). Target cells were incubated with titrated concentrations of mAbs for 10 min and then with primary human NK cells at an effector:target ratio of 9:1. ADCC was measured using LDH release assay (Cytoxicity Detection Kit (LDH) (Roche; cat. no. 11644793001) after 4 h incubation at 37˚C.

ADCP assays were performed using ExpiCHO-S cells transiently transfected with SARS-CoV-2 spike glycoproteins (Wuhan D614, BA.1, or BA.2) and labelled

with PKH67 (Sigma Aldrich) as targets. PMBCs from healthy donors were labelled with CellTrace Violet (Invitrogen) and used as source of phagocytic effector cells. Target cells (1000 per well) were incubated with titrated concentrations of mAbs for 10 min and then mixed with PBMCs (200,000 per well). The next day, cells were stained with APC-labelled anti-CD14 mAb (BD Pharmingen), BV605-labelled anti-CD16 mAb (BioLegend), BV711-labelled anti-CD19 mAb (BioLegend), PerCP/Cy5.5-labelled anti-CD3 mAb (BioLegend), APC/Cy7-labelled anti-CD56 mAb (BioLegend) for the identification of CD14+ monocytes. After 20 min, cells were washed and fixed with 4% paraformaldehyde before acquisition on a ZE5 Cell Analyzer (Bio-Rad). Data were analyzed using FlowJo v10 software. The % ADCP was calculated as % of monocytes (CD3- CD19- CD14+ cells) positive for PKH67.

**MAb affinity measurements**. (a) S309. SARS-CoV-2 RBD constructs contain residues 328–531 of the spike protein from GenBank NC_045512.2 with an N-terminal signal peptide and a C-terminal 8xHis-AviTag. Proteins were expressed in Expi293F cells (Thermo Fisher Scientific) at 37 °C and 8% $CO_2$. Transfections were performed using the ExpiFectamine 293 Transfection Kit (Thermo Fisher Scientific). Cell culture supernatants were harvested five days after transfection by spinning at $4,000 \times g$ for 20 min. Supernatants were then filtered through a 0.22 µm filter and supplemented with 10× PBS to a final concentration of 2.5× PBS (342.5 mM NaCl, 6.75 mM KCl and 29.75 mM phosphates). SARS-CoV-2 RBDs were purified using HisPur Cobalt resin (Thermo Fisher Scientific) followed by buffer exchange into PBS using Amicon centrifugal filters (MilliporeSigma). S309 Fab binding measurements using surface plasmon resonance were performed on a Biacore T200 instrument. A CM5 chip with covalently immobilized anti-Avi polyclonal antibody (GenScript, Cat #: A00674-40) was used for surface capture of His-Avi tag containing RBDs. Running buffer was HBS-EP + pH 7.4 (Cytiva) and measurements were performed at 25 °C. Experiments were performed with a 4-fold dilution series of monomeric S309 Fab: 571, 143, and 36 nM and were run as single-cycle kinetics. Data were double reference-subtracted and fit to a binding model using Biacore Evaluation software. The 1:1 binding model was used to estimate the kinetics parameters. The experiment was performed in triplicate with technical replicates for each ligand (RBDs). Kinetics values out of instrument's limit were omitted. $K_D$ values were reported as the average of all replicates.

(b) AZD8895 or AZD1061. Purified SARS-CoV-2 BA.2 RBD protein was purchased from AcroBiosystem. SARS-CoV-2 RBD constructs for D614G (residues 334–526), BA.1, and BA.1.1 (residues 319–537) were cloned with an N-terminal CD33 leader sequence and C-terminal GSSG linker, AviTag, GSSG linker, and 8xHisTag. Recombinant proteins were expressed in FreeStyle 293X or 293 F cells (Thermo Fisher Scientific) following transfection with 293fectin (Thermo Fisher Scientific) according to manufacturer's directions. RBD proteins were isolated by affinity chromatography using a HisTrap column (GE Healthcare), followed by size exclusion column chromatography (GE Healthcare). AZD8895 and AZD1061 mAbs were isolated from expression supernatants by affinity chromatography using either MabSelect, protein A or protein G columns (GE Healthcare) and eluted with 0.1 M glycine at low pH. MAbs were dialyzed into PBS. Fab fragments were prepared using papain immobilized on agarose resin (Thermo Fisher) according to manufacturer's directions, followed by purification over either MabSelect or protein A columns (GE Healthcare/Cytiva).Purified proteins were pooled and analyzed by gel electrophoresis (SDS-PAGE) to ensure purity and appropriate molecular weights, as well as further evaluation by size-exclusion chromatography coupled to multi-angle light scattering for some proteins.

Kinetic rates ($k_a$, $k_d$, and $K_D$) of AZD1061 and AZD8895 binding to SARS-CoV-2 RBD protein were evaluated by biolayer interferometry using an Octet model RED384 instrument (FortéBio). Anti-His (HIS1K) biosensors (Pall FortéBio/Sartorius Part#18–5120) tips were first soaked in 1x kinetics buffer for 10 min followed by a baseline signal measurement in 1x kinetics buffer for 60 sec. His-tagged SARS-CoV-2 spike RBD diluted in kinetics buffer to 5 µg/mL was loaded onto the tips for 180 sec. Antigen containing tips were then added to wells containing serial dilutions of AZD8895 or AZD1061 (at concentrations ranging from 3000 nM to 200 nM). Association and dissociation measurements were made for 300 sec each. All steps were performed at 30 °C and 1000 rpm shaking. Data were reference subtracted and fit to a 1:1 binding model using Octet Data Analysis Software 12.0. The fitted data were plotted with GraphPad Prism software (version 9.0).

**Cytokine and Chemokine protein measurements**. Lung homogenates were incubated with Triton-X-100 (1% final concentration) for 1 h at room temperature to inactivate SARS-CoV-2. Homogenates were analyzed for cytokines and chemokines by Eve Technologies Corporation (Calgary, AB, Canada) using their Mouse Cytokine Array/Chemokine Array 31-Plex (MD31) platform.

**Lung pathology**. Animals were euthanized before harvest and fixation of tissues. Briefly, lungs were inflated with approximately 1.2 mL of 4% paraformaldehyde using a 3 mL syringe and catheter inserted into the trachea. Tissues were allowed to fix for 24 h at room temperature, embedded in paraffin, and sections were stained with hematoxylin and eosin. Slides were scanned using a Hamamatsu NanoZoomer slide scanning system, and the images were viewed using NDP view software (ver.1.2.46).

**Statistical analysis**. All statistical tests were performed as described in the indicated figure legends using Prism v8.0 or 9.0. Statistical significance was determined using a one-way ANOVA when comparing three or more groups. When comparing two groups, a Mann-Whitney test was performed. The number of independent experiments performed are indicated in the relevant figure legends. For correlation analyses, best-fit lines were calculated using a simple linear regression. Two-tailed Pearson correlation was used to calculate the $R^2$ and P values indicated within each panel.

**Reporting summary**. Further information on research design is available in the Nature Research Reporting Summary linked to this article.

## Data availability
All data supporting the findings of this study are available within the paper, in the Source Data, and from the corresponding author upon request. Source data are provided as Source Data files. There are no restrictions in obtaining access to primary data. Models of mAb complexes were generated from their respective PDB files with the following accession codes: COV2–2196 (AZD8895; PDB: 7L7D); COV2–2130 (AZD1061; PDB: 7L7E); S309 (PDB: 6WPS). Source data are provided with this paper.

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

## Acknowledgements
This study was supported by grants and contracts from the NIH (R01 AI157155, U01 AI151810, NIAID Centers of Excellence for Influenza Research and Response (CEIRR) contract 75N93019C00051) and the Defense Advanced Research Projects Agency (DARPA; HR0011-18-2-0001). J.B.C. is supported by a Helen Hay Whitney Foundation postdoctoral fellowship. E.A.M. is supported by a W.M. Keck postdoctoral fellowship from Washington University. We thank Gloria Lombardo and Selina Feller for technical support and Abigail Powell, Josh Dillen, and Nadine Czudnochowski for protein production assistance.

## Author contributions

J.B.C. performed and analyzed neutralization assays. J.M.E. performed structural analyses with guidance from D.H.F. J.B.C., S.M., Z.C., B.W., and E.A.M. performed mouse experiments and viral burden analyses. J.B.C. propagated and validated SARS-CoV-2 viruses. B.G. and M.A.S. designed, performed, and analyzed in vitro Fc-mediated effector function studies. K. Rosenthal, K. Ren, G.S., and H.V.D. performed binding affinity measurements. A.J., L.D., and S.A.H. performed deep sequencing analysis. L.A.P., D.C., Y-M.L., and M.T.E. provided mAbs. P.J.H. and Y.K. provided SARS-CoV-2 strains. J.E.C. and H.W.V. provided key intellectual contributions to the design of the study. D.H.F. and M.S.D. obtained funding and supervised the research. J.B.C. and M.S.D. wrote the initial draft, with the other authors providing editorial comments.

## Competing interests

M.S.D. is a consultant for Inbios, Vir Biotechnology, Senda Biosciences, and Carnival Corporation, and on the Scientific Advisory Boards of Moderna and Immunome. The Diamond laboratory has received funding support in sponsored research agreements from Moderna, Vir Biotechnology, and Emergent BioSolutions. J.E.C. has served as a consultant for Luna Innovations, Merck, and GlaxoSmithKline, is a member of the Scientific Advisory Board of Meissa Vaccines and is founder of IDBiologics. The Crowe laboratory has received sponsored research agreements from AstraZeneca, Takeda, and IDBiologics during the conduct of the study. Vanderbilt University has applied for patents for some of the antibodies in this paper, for which J.E.C. is an inventor. B.G., M.A.S, G.S., H.V.D., H.W.V., D.C., and L.A.P. are employees of Vir Biotechnology and may hold equity in Vir Biotechnology. L.A.P. is a former employee and may hold equity in Regeneron Pharmaceuticals. H.W.V. is a founder and holds shares in PierianDx and Casma Therapeutics. Neither company provided resources to this study. Y.K. has received unrelated funding support from Daiichi Sankyo Pharmaceutical, Toyama Chemical, Tauns Laboratories, Inc., Shionogi & Co. LTD, Otsuka Pharmaceutical, KM Biologics, Kyoritsu Seiyaku, Shinya Corporation, and Fuji Rebio. K. Rosenthal, K. Ren, Y-M.L. and M.T.E. are employees of AstraZeneca and may hold stock in AstraZeneca. All other authors declare no competing interests.
