## [Peer Review File · Nature Communications]

Reviewer comments, first round –

Reviewer #1 (Remarks to the Author):

The RBD is the major target for SARS-CoV-2 neutralizing nanobodies that are in advanced development or that received approval for clinical usage to treat SARS-CoV-2 infections. However, since the emerge of especially the Omicron variants (BA.1, BA.1.1 and BA.2) most of these antibodies showed dramatic loss in potency to at least one of these variants. This is also the case for Sotrovimab (an antibody based on the SARS-CoV-1 neutralizing S309 human monoclonal) and for the 2 Evusheld (AZD7442) antibodies. In the submitted manuscript Case demonstrate that albeit substantial reduction in neutralizing activity in vitro prophylactic treatment using these antibodies can still provide significant protection against pulmonary infection of K18-hACE2 mice with these variants. Whereas the Sotrovimab has fully active Fc effector functions those of Evusheld are reduced due to the engineered TM substitution. In correspondence with this the authors demonstrate that the observed in vivo protection by Evusheld for the variants of concern directly correlates with its in vitro neutralization activity for these variants. In sharp contrast prophylactic treatment with Sotrovimab provides protection against infection with SARS-CoV-2 variants (including the early 614G strain) via it's Fc effector functions. This finding underscores the importance of IgG Fc effector functions for at least some antibodies and argues that these should be taken into consideration to evaluate the protective activity of these antibodies. The presented manuscript is highly relevant for the field, the experiments are well designed and described and the results clearly support the conclusions made by the authors. I have only one few experimental remarks and some suggestions to present the data in a clearer way.

Major remark

1. Although Sotrovimab and Evusheld are used to therapeutically treat infected patients, only prophylactic treatment has been investigated in the manuscript. This has been noted by the authors as a limitation. One of the authors has previously demonstrated that MPE8, a broadly RSV neutralizing antibody requires its Fc effector function for in vivo protection when used in a prophylactic setting but not when used in a therapeutic setting (Corte et al, Nature 2013). As both Sotrovimab and Evushelds are used as therapeutic treatment it is important to test the protective activity of these antibodies for the Omicron variants in a therapeutic setting and the role of Fc effector functions of Sotrovimab herein.

Minor remarks

1. Was the neutralizing activity of the AZD7442TM variant (without YTE) that was used for the in vivo experiments confirmed?
2. Similar to the table in fig 1, summarizing the neutralization activity for the viral variants a table summarizing the fold reduction in viral lung RNA for the challenge experiments would be very helpful for comparison. (eg as a supplementary table).
3. In Figure 2 k and l, the fold reduction in neutralizing activity of AZD7442(TM) and S309-LS is plotted against the fold change in lung viral titer. As for most treated animals the lung viral titer is near or at the limit of quantification the latter rather reflects the level of viral replication of the variants in isotype control IgG treated animals. Moreover, titration of lung viral titers might be affected by ex-vivo neutralization. Therefore, please use the fold change in lung viral RNA instead of fold change the viral titer in these plots
4. Figure 3b shows a series of histograms to illustrate that antibody binding to cells expressing spikes of omicron variants is affected to some extent as compared to original spikes. Differences in the binding of antibodies to cells expressing the spikes of the variants would be more clearly illustrated when shown as binding curves (MFI vs IgG concentration).
5. Figure 3.e demonstrates that VIR-7831 can instruct ADCP of cells expressing the BA.2 spike. However, the plateau that is reached for BA.2 is lower than that for BA.1 and Wu-D614. Is this difference in plateau significant?
6. Similar to the plots shown in figure 2.k plots displaying the fold reduction in lung viral RNA vs fold reduction in IgG binding to spike expressing cells or the ADCC (AUC) or the ADPC (AUC or plateau) would be very informative (eg as supplementary fig)
7. In the binding, ADCC and ADCP assays the spike of Wu-D614 is used whereas the neutralization experiments an in vivo challenges have been performed using the D614G variant. Please clarify

why.

8. In the challenge experiment a dose of 10mg/kg is used. Please indicate in the text the dosage that is used to treat patients.

9. Please clarify why 3mg/kg dosage was used for the huFcγR challenge and not 10 mg/kg as in the other challenge experiments.

Bert Schepens

Reviewer #2 (Remarks to the Author):

The manuscript by Case et al. evaluated two mAb therapeutics (S309 and AZD7442: AZD8895+AZD1061) for neutralization in vitro and protection of infection against three Omicron variants (BA.1, BA.1.1, and BA.2) in mouse model. In comparison with neutralizing activity against the D614G virus, both S309 and AZD7442 showed some level of reduced neutralizing activity as shown in EC50s. In mouse model study, prophylactic treatment with the antibodies showed some levels of reduction of viral load in lung but not in upper respiratory system. The authors also evaluated Fc-mediated ADCC and ADCP functions and showed Fc mediated effector function as a mechanism of action for S309 mAb. Overall, the findings in the study are not surprising because epitopes of the monoclonal antibodies often can predict the efficacy loss based on many mutations in Omicron variants. Since Omicron variants currently are still dominant in COVID-19 spreading in the world, this study evaluated two available antibodies with Emergency Use Authorization (EUA) status against Omicron variants and provided a valuable data set for selection of clinical treatment options. Overall, study is well designed and data support the conclusion.

Comments to authors:

1. The in vivo efficacy of mAbs was evaluated as prophylaxis and efficacy for therapeutic efficacy (after viral infection) will be important data to be added for comparison.
2. The viral titers were examined on days 6/7 after infection, examining viral titers on dpi 2/3 would provide some pharmacokinetics for the antibody treatment.
3. S309-LS treatment reduced pro-inflammatory cytokines and chemokines compared to isotype, as shown in Fig. 2i-j. Authors showed that Fc mediated effector functions served as a mechanism of action in S309-LS but not S309-GRLR. The comparison of cytokine/chemokine profiles of S309 with GRLR would be interesting data to be added to dissect the profiles induced by viral infection and activation of immune effector function.

Minor comments:

1. Fig. 3b, it would be helpful to plot MFI vs antibody concentrations in a graph to see compare the two antibody bindings.
2. Fig.3c-f, the ADCC killing (%) is only 10% range and need to include isotype control IgG for spontaneous killing in the assay.
3. Ref 15 and 16 are the same one?

Reviewer #3 (Remarks to the Author):

Case et al studied SARS-CoV-2 mabs for prevention of infection in mice against Omicron variants. They show the S309 (Sotrovimab) mab loses in vitro activity against BA2 but retains some protective efficacy, in part related to Fc functions. The work is well done and of interest to the field, although in my view further animal work to discern clinical relevance is needed.

Comments

1. The mouse experiments were all performed in prevention setting, whereas most mab use clinically is in a treatment setting. It would be useful to assess efficacy post infection. A dose titration should be studied in mice – I note that Sotrovimab is being studied in higher doses now clinically, in line with recent modelling work (<https://www.medrxiv.org/content/10.1101/2022.03.21.22272672v1>). Previous work has

suggested F_c functions are more important in treatment than prevention settings. Although noted in the limitations, I think such work would be relevant given the clinical implications.

2. For unclear reasons, mouse infections can be less susceptible to strain variability than larger animal models or humans. E.g. see

<https://www.biorxiv.org/content/10.1101/2022.02.07.479468v1>

Although I appreciate the difficulties, given the clinical implications of this work I think a second model would be useful.

RESPONSE TO REVIEWER COMMENTS

Reviewer #1

The RBD is the major target for SARS-CoV-2 neutralizing nanobodies that are in advanced development or that received approval for clinical usage to treat SARS-CoV-2 infections. However, since the emerge of especially the Omicron variants (BA.1, BA.1.1 and BA.2) most of these antibodies showed dramatic loss in potency to at least one of these variants. This is also the case for Sotrovimab (an antibody based on the SARS-CoV-1 neutralizing S309 human monoclonal) and for the 2 Evusheld (AZD7442) antibodies. In the submitted manuscript Case demonstrate that albeit substantial reduction in neutralizing activity in vitro prophylactic treatment using these antibodies can still provide significant protection against pulmonary infection of K18-hACE2 mice with these variants. Whereas the Sotrovimab has fully active Fc effector functions those of Evusheld are reduced due to the engineered TM substitution. In correspondence with this the authors demonstrate that the observed in vivo protection by Evusheld for the variants of concern directly correlates with its in vitro neutralization activity for these variants. In sharp contrast prophylactic treatment with Sotrovimab provides protection against infection with SARS-CoV-2 variants (including the early 614G strain) via it's Fc effector functions. This finding underscores the importance of IgG Fc effector functions for at least some antibodies and argues that these should be taken into consideration to evaluate the protective activity of these antibodies.

The presented manuscript is highly relevant for the field, the experiments are well designed and described and the results clearly support the conclusions made by the authors. I have only one few experimental remarks and some suggestions to present the data in a clearer way.

We greatly appreciate the favorable summary of the paper.

Major remark

1. Although Sotrovimab and Evusheld are used to therapeutically treat infected patients, only prophylactic treatment has been investigated in the manuscript. This has been noted by the authors as a limitation. One of the authors has previously demonstrated that MPE8, a broadly RSV neutralizing antibody requires its Fc effector function for in vivo protection when used in a prophylactic setting but not when used in a therapeutic setting (Corte et al. Nature 2013). As both Sotrovimab and Evushelds are used as therapeutic treatment it is important to test the protective activity of these antibodies for the Omicron variants in a therapeutic setting and the role of Fc effector functions of Sotrovimab herein.

*We agree with the reviewer's comment. In the revision, we have added post-exposure therapeutic challenge data with S309-LS (sotrovimab) and AZD7442-TM (tixagevimab and cilgavimab: Evusheld) against all of the Omicron variants. These data (new **Figure 4g-i**) show protective activity of the mAbs that is similar in pattern to the prophylaxis data. For S309-LS, despite losing substantial neutralizing activity against BA.2, protection was still observed, suggesting that Fc effector functions have separate roles in the control of SARS-CoV-2 infection. Analogously, other studies (Bournazos et al., 2020) have shown that in both prophylactic and therapeutic settings, Fc engineering of anti-influenza IgG mAbs for selective binding to the activating FcγR, FcγRIIa results in enhanced protective efficacy against infection.*

Minor remarks

1. Was the neutralizing activity of the AZD7442TM variant (without YTE) that was used for the in vivo experiments confirmed?

Yes, we have confirmed there is no difference in neutralizing activity of AZD7442-TM (without YTE) against Omicron variants. Some of these data were published in a prior study (VanBlargan et al., 2022). We have added a comment on this point (p. 6).

2. Similar to the table in fig 1, summarizing the neutralization activity for the viral variants a table summarizing the fold reduction in viral lung RNA for the challenge experiments would be very helpful for comparison. (eg as a supplementary table).

*We agree and have created a new **Supplementary Table S1**, which summarizes these data.*

3. In Figure 2 k and l, the fold reduction in neutralizing activity of AZD7442(TM) and S309-LS is plotted against the fold change in lung viral titer. As for most treated animals the lung viral titer is near or at the limit of quantification the latter rather reflects the level of viral replication of the variants in isotype control IgG treated animals. Moreover, titration of lung viral titers might be affected by ex-vivo neutralization. Therefore, please use the fold change in lung viral RNA instead of fold change the viral titer in these plots.

*We actually did use the fold change viral RNA levels on the y-axis for the reasons outlined by the Reviewer. This unit was indicated in the figure legend. That said, we have further qualified this point in the main text (p. 8) and on the y-axis of **Figure 2 k and l**.*

4. Figure 3b shows a series of histograms to illustrate that antibody binding to cells expressing spikes of omicron variants is affected to some extent as compared to original spikes. Differences in the binding of antibodies to cells expressing the spikes of the variants would be more clearly illustrated when shown as binding curves (MFI vs IgG concentration).

*We agree with this comment, and have added a graph of these data to **Figure 3c** as suggested.*

5. Figure 3.e demonstrates that VIR-7831 can instruct ADCP of cells expressing the BA.2 spike. However, the plateau that is reached for BA.2 is lower than that for BA.1 and Wu-D614. Is this difference in plateau significant?

*The differences in ADCP for VIR-7831 between each of the tested variants (Wu-D614, BA.1, and BA.2) are not statistically significant. Rather, the differences between VIR-7831 and the GRLR versions, including for the BA.2 variant, are significant and are now indicated in **Figure 3g**.*

6. Similar to the plots shown in figure 2.k plots displaying the fold reduction in lung viral RNA vs fold reduction in IgG binding to spike expressing cells or the ADCC (AUC) or the ADPC (AUC or plateau) would be very informative (eg as supplementary fig).

*We agree and have calculated the correlation between the fold-change in EC_{50} value and the fold-change in Fab affinity. We have included these data as **Extended Data Figure 3**.*

7. In the binding, ADCC and ADCP assays the spike of Wu-D614 is used whereas the neutralization experiments and in vivo challenges have been performed using the D614G variant. Please clarify why.

The ADCC and ADCP assays were performed in the Corti laboratory using a Wu-D614 spike protein. All live virus assays used a D614G version in the Diamond laboratory. The antibodies tested, S309-LS (sotrovimab) and AZD7442-TM (tixagevimab and cilgavimab), bind to the RBD at sites far from the D614 residue. Accordingly, we have not seen differences previously with D614 or D614G.

8. In the challenge experiment a dose of 10mg/kg is used. Please indicate in the text the dosage that is used to treat patients.

In the text, we have added a comment on the clinical dosing of the antibodies used (p. 6).

9. Please clarify why 3 mg/kg dosage was used for the huFcγR challenge and not 10 mg/kg as in the other challenge experiments.

These studies are performed in a different model (human FcγR transgenic mice) with a different SARS-CoV-2 variant (B.1.351) using a different inoculation dose. In separate studies, we have optimized S309 mAb protection in this model and showed that 3 mg/kg is sufficient for 100- to 1,000-fold reductions in viral RNA levels.

Reviewer #2

The manuscript by Case et al. evaluated two mAb therapeutics (S309 and AZD7442: AZD8895+AZD1061) for neutralization in vitro and protection of infection against three Omicron variants (BA.1, BA.1.1, and BA.2) in mouse model. In comparison with neutralizing activity against the D614G virus, both S309 and AZD7442 showed some level of reduced neutralizing activity as shown in EC50s. In mouse model study, prophylactic treatment with the antibodies showed some levels of reduction of viral load in lung but not in upper respiratory system. The authors also evaluated Fc-mediated ADCC and ADCP functions and showed Fc mediated effector function as a mechanism of action for S309 mAb. Overall, the findings in the study are not surprising because epitopes of the monoclonal antibodies often can predict the efficacy loss based on many mutations in Omicron variants. Since Omicron variants currently are still dominant in COVID-19 spreading in the world, this study evaluated two available antibodies with Emergency Use Authorization (EUA) status against Omicron variants and provided a valuable data set for selection of clinical treatment options. Overall, study is well designed and data support the conclusion.

We appreciate the supportive comments.

Comments to authors:

1. The in vivo efficacy of mAbs was evaluated as prophylaxis and efficacy for therapeutic efficacy (after viral infection) will be important data to be added for comparison.

*We agree with this comment. In the revision, we have added post-exposure therapeutic challenge data with S309-LS (sotrovimab) and AZD7442-TM (tixagevimab and cilgavimab) against all of the Omicron variants. These data (new **Figure 4g-i**) show protective activity of the mAbs that is similar in pattern to the prophylaxis data. For S309-LS, despite losing substantial neutralizing activity against BA.2, protection was still observed.*

2. The viral titers were examined on days 6/7 after infection. Examining viral titers on dpi 2/3 would provide some pharmacokinetics for the antibody treatment.

*As the reviewer can appreciate, these studies are complex (multiple mAbs, multiple virus strains), quite resource-intensive, and already have used large numbers of mice (>300) under A-BSL3 conditions. Respectfully, while this is an interesting experiment, we do not feel that it will substantially change the conclusions of the paper. We have evaluated the pharmacokinetics of these mAbs in mice and do not see large reductions over one week of the experiment (see **Reviewer Figure R1** below).*

Reviewer Figure R1. Pharmacokinetic analyses of S309-LS and AZD7442-TM in C57BL/6 mice. Mice were treated with 200 µg of S309-LS or AZD7442-TM. On the indicated day post-treatment, sera, lungs, and nasal washes were collected, and the concentration of human antibody present in different mouse tissues was determined by ELISA (n = 3 per mAb per time point).

3. S309-LS treatment reduced pro-inflammatory cytokines and chemokines compared to isotype, as shown in Fig. 2i-i. Authors showed that Fc mediated effector functions served as a mechanism of action in S309-LS but not S309-GRLR. The comparison of cytokine/chemokine profiles of S309 with GRLR would be interesting data to be added to dissect the profiles induced by viral infection and activation of immune effector function.

We agree this is an interesting experiment. In the revision, we have added these data (Figure 4d and Extended Data Figure 9) showing that infected mice treated with AZD7442-TM or S309-LS mAbs have reduced levels of pro-inflammatory cytokines and chemokines (except for AZD7442-TM against BA.1.1).

Minor comments:

1. Fig. 3b, it would be helpful to plot MFI vs antibody concentrations in a graph to see compare the two antibody bindings.

We agree with this comment, and have added a graph showing these data to Figure 3c as suggested.

2. Fig.3c-f, the ADCC killing (%) is only 10% range and need to include isotype control IgG for spontaneous killing in the assay.

In our ADCC killing experiments, we did not include an isotype control mAb. Rather, we used GRLR-matched mAbs that allowed us to measure the Fc dependence of the effect for each antibody compared to cell-only controls. The GRLR mAb treatments produced identical results to the cell-only controls and are in agreement with negative controls for other mAbs in the field (Hansen et al., 2020).

3. Ref 15 and 16 are the same one?

Yes, this was an error. We have corrected this mistake.

Reviewer #3

Case et al studied SARS-CoV-2 mAbs for prevention of infection in mice against Omicron variants. They show the S309 (Sotrovimab) mAb loses in vitro activity against BA2 but retains some protective efficacy, in part related to Fc functions. The work is well done and of interest to the field, although in my view further animal work to discern clinical relevance is needed.

We agree with this comment.

Comments

1. The mouse experiments were all performed in prevention setting, whereas most mAb use clinically is in a treatment setting. It would be useful to assess efficacy post infection. A dose titration should be studied in mice – I note that Sotrovimab is being studied in higher doses now clinically, in line with recent modelling work (<https://www.medrxiv.org/content/10.1101/2022.03.21.22272672v1>). Previous work has suggested Fc functions are more important in treatment than prevention settings. Although noted in the limitations, I think such work would be relevant given the clinical implications.

*We agree with the comment. In the revision, we have added post-exposure therapeutic challenge data with S309-LS (sotrovimab) and AZD7442-TM (tixagevimab and cilgavimab) against all of the Omicron variants. These data (new **Figure 4g-i**) show protective activity of the mAbs that is similar in pattern to the prophylaxis data. For S309-LS, despite losing substantial neutralizing activity against BA.2, protection was still observed.*

2. For unclear reasons, mouse infections can be less susceptible to strain variability than larger animal models or humans. E.g. see <https://www.biorxiv.org/content/10.1101/2022.02.07.479468v1>. Although I appreciate the difficulties, given the clinical implications of this work I think a second model would be useful.

We are uncertain of the exact issue being highlighted by the reviewer. Indeed, our group includes authors on the cited BioRxiv paper, which evaluates serum antibody responses to vaccination or infection in mice and shows how they differentially neutralize variants (compared to humans) – unless, we are missing something, this induced Ab response data does not seem directly related to the mAb passive transfer studies we conducted here. While additional animal models with Omicron variants would be interesting, the viruses also are attenuated in hamsters and non-human primates (Gagne et al., 2022; Halfmann et al., 2022). After discussion with the Editor, we feel that such studies are beyond the scope of this manuscript.

LITERATURE CITED.

Bournazos, S., Corti, D., Virgin, H.W., and Ravetch, J.V. (2020). Fc-optimized antibodies elicit CD8 immunity to viral respiratory infection. *Nature*. 588(7838):485-490.

Gagne, M., Moliva, J.I., Foulds, K.E., Andrew, S.F., Flynn, B.J., Werner, A.P., Wagner, D.A., Teng, I.T., Lin, B.C., Moore, C., *et al.* (2022). mRNA-1273 or mRNA-Omicron boost in vaccinated macaques elicits similar B cell expansion, neutralizing responses, and protection from Omicron. *Cell* 185, 1556-1571.e1518.

Halfmann, P.J., Iida, S., Iwatsuki-Horimoto, K., Maemura, T., Kiso, M., Scheaffer, S.M., Darling, T.L., Joshi, A., Loeber, S., Singh, G., *et al.* (2022). SARS-CoV-2 Omicron virus causes attenuated disease in mice and hamsters. *Nature*.

Hansen, J., Baum, A., Pascal, K.E., Russo, V., Giordano, S., Wloga, E., Fulton, B.O., Yan, Y., Koon, K., Patel, K., *et al.* (2020). Studies in humanized mice and convalescent humans yield a SARS-CoV-2 antibody cocktail. *Science* 369, 1010-1014.

VanBlargan, L.A., Errico, J.M., Halfmann, P.J., Zost, S.J., Crowe, J.E., Jr., Purcell, L.A., Kawaoka, Y., Corti, D., Fremont, D.H., and Diamond, M.S. (2022). An infectious SARS-CoV-2 B.1.1.529 Omicron virus escapes neutralization by therapeutic monoclonal antibodies. *Nat Med*, 1-6.

Reviewer comments, second round –

Reviewer #1 (Remarks to the Author):

The authors took into account my remarks adequately and adapted the manuscript accordingly. In my opinion this very nice manuscript is eligible for publication at Nature Communications.

Reviewer #2 (Remarks to the Author):

The revised manuscript addressed some of my comments and concerns, but Authors should discuss how the antibody S309 Fc mediated ADCC/ADCP activities (measured in vitro, Fig. 3) impact the reduced levels of inflammatory cytokines in lung tissues treated by S309-LS. Since ADCC/ADCP can activate immune effector cells such as NK cells and macrophages and lead to increased production of cytokines such as TNF- α and IFN- γ as well documented in cancer therapeutic antibodies targeting tumor associated antigens. Reduction of viral load clearly can attribute to the reduced inflammatory cytokines, but it seems counterintuitive for ADCC/ADCP activity in connection with the cytokine reduction.

Reviewer #3 (Remarks to the Author):

I am satisfied with the response

REVIEWERS' COMMENTS

Reviewer #1:

The authors took into account my remarks adequately and adapted the manuscript accordingly. In my opinion, this very nice manuscript is eligible for publication at Nature Communications.

We thank the reviewer for their positive comments.

Reviewer #2:

The revised manuscript addressed some of my comments and concerns, but Authors should discuss how the antibody S309 Fc mediated ADCC/ADCP activities (measured in vitro, Fig. 3) impact the reduced levels of inflammatory cytokines in lung tissues treated by S309-LS. Since ADCC/ADCP can activate immune effector cells such as NK cells and macrophages and lead to increased production of cytokines such as TNF- α and IFN- γ as well documented in cancer therapeutic antibodies targeting tumor associated antigens. Reduction of viral load clearly can attribute to the reduced inflammatory cytokines, but it seems counterintuitive for ADCC/ADCP activity in connection with the cytokine reduction.

The reviewer raises an interesting point. We speculate that S309-LS Fc-mediated protection through ADCC/ADCP did not show a pro-inflammatory cytokine signature because of the timepoint that we used for analysis. Indeed, we have observed that pro-inflammatory responses to SARS-CoV-2 infection in K18-hACE2 mice peak at approximately 6-7 dpi (the timepoint used here). However, at this timepoint, since S309-LS was present before viral inoculation (administered at day -1), Fc γ R engagement and recruitment of immune effector cells might be resolved by day +7, and the lower cytokine levels seem reflect the decrease in viral infection. To fully understand the immune effector cells, anatomical locale, and timing of the Fc γ R-mediated protection of S309-LS and the cytokine responses thereof, future kinetic studies are needed. We discuss this caveat in the limitations of study section.

Reviewer #3:

I am satisfied with the response

We thank the reviewer for their comment.